# Endovascular Bariatric Surgery as Novel Minimally Invasive Technique for Weight Management in the Morbidly Obese: Review of the Literature

**DOI:** 10.3390/nu13082541

**Published:** 2021-07-25

**Authors:** Giuseppe Massimo Sangiorgi, Alberto Cereda, Nicola Porchetta, Daniela Benedetto, Andrea Matteucci, Michela Bonanni, Gaetano Chiricolo, Antonino De Lorenzo

**Affiliations:** 1Department of Biomedicine and Prevention, Institute of Cardiology, Cardiac Cath Lab, University of Rome Tor Vergata, 00133 Rome, Italy; nicola.porchetta@gmail.com (N.P.); dania.benedetto@gmail.com (D.B.); andrea.matteucci2@gmail.com (A.M.); michelabonanni91@gmail.com (M.B.); gaetano.chiricolo@ptvonline.it (G.C.); delorenzo@uniroma2.it (A.D.L.); 2Department of Cardiology, Cardiac Cath Lab, San Gaudenzio Clinic, 28100 Novara, Italy; tskcer@hotmail.it

**Keywords:** bariatric surgery, cardiovascular disease, endovascular bariatric surgery, obesity, prevention

## Abstract

Nowadays, obesity represents one of the most unresolved global pandemics, posing a critical health issue in developed countries. According to the World Health Organization, its prevalence has tripled since 1975, reaching a prevalence of 13% of the world population in 2016. Indeed, as obesity increases worldwide, novel strategies to fight this condition are of the utmost importance to reduce obese-related morbidity and overall mortality related to its complications. Early experimental and initial clinical data have suggested that endovascular bariatric surgery (EBS) may be a promising technique to reduce weight and hormonal imbalance in the obese population. Compared to open bariatric surgery and minimally invasive surgery (MIS), EBS is much less invasive, well tolerated, with a shorter recovery time, and is probably cost-saving. However, there are still several technical aspects to investigate before EBS can be routinely offered to all obese patients. Further prospective studies and eventually a randomized trial comparing open bariatric surgery vs. EBS are needed, powered for clinically relevant outcomes, and with adequate follow-up. Yet, EBS may already appear as an appealing alternative treatment for weight management and cardiovascular prevention in morbidly obese patients at high surgical risk.

## 1. Introduction

Nowadays, obesity represents one of the most unresolved global pandemics, posing a critical health issue in developed countries. According to the World Health Organization, its prevalence has tripled since 1975, reaching 13% of the world population in 2016. Obesity is defined as a body mass index (BMI) greater than 30 kg/m^2^, while morbid obesity is defined as BMI > 40 kg/m^2^. Numerous comorbidities such as major stroke, acute myocardial infarction, hypertension, type 2 diabetes, hyperlipidemia, obstructive sleep apnea, and all-cause mortality are strongly associated with this disease [1]. As a result, approximately 2.8 million deaths per year may occur in adult populations affected by obesity. 

Given the tremendous impact on public health with an approximate attributable cost of nearly 150 billion dollars per year in the United States, obesity treatments have become of critical importance for the healthcare system, the medical community, and policymakers [2].

The cornerstone of obesity treatment is represented by behavioral modifications (i.e., diet and physical exercise), ideally in a highly motivated patient that should be followed by a multidisciplinary team of healthcare professionals. If successful, this strategy consents modest and durable weight loss reduction of 5% to 10% [3]. The long-term efficacy of all behavioral therapies is limited. For those unable to reach these goals, few drugs (orlistat, lorcaserin, phentermine/topiramate, nartrexone/bupropion, semaglutide and liraglutide) are available as adjuvant therapy, but in general, are not free of side effects, usually dose-dependent, have limited adherence (frequently due to arbitrary withdrawal of the drug), and with suboptimal outcome in obtaining the goal of weight-reduction. Recently, non-surgical endoscopic bariatric therapies such as intragastric balloons, endoscopic gastric plication, and endoluminal duodenal-jejunal sleeve have been implemented in patients not willing to undergo conventional bariatric surgery. However, potentially severe complications have been reported with these techniques (gastric perforation, bowel obstruction, and gastrointestinal bleeding), and for these reasons they are currently performed only in highly experienced centers. 

Surgical approaches, mainly represented by Roux-en-Y gastric bypass, adjustable gastric banding, sleeve gastrectomy, and biliopancreatic division (Figure 1) are reserved to morbidly obese individuals or obese individuals with one or more obesity-related comorbidities (or even lower for uncontrolled diabetes) who have not been able to reach the aforementioned goals with behavioral modifications and drug therapy. 

Well-known short- and long-term complications, even if uncommon, including bleeding, infections, deep venous thrombosis, gastric dumping syndrome, and internal hernia have been reported with different surgical techniques. 

Another frequent eventuality is post-bariatric surgery anemia: it is in most cases due to iron deficiency, along with vitamin B12 deficiency as a secondary cause. Iron deficiency is expressed by low serum ferritin and it occurs because of its lower absorption secondary to hypocloridria and the bypassing of the duodenum and proximal jejunum [4]. In addition to anemia, vitamin B12 deficiency (resulting from inadequate secretion of intrinsic factor, limited gastric acidity and the bypassing of the duodenum) can lead to neurological disorders [4]. In the absence of adequate vitamin B12 supplement, up to 30% of patients will be unable to maintain normal levels of plasma B12 at 1 year [5]. 

Bariatric surgery results in calcium/vitamin D malabsorption (results from bypassing the duodenum and proximal jejunum) with secondary hyperparathyroidism, changes in fat mass and alterations in fat- and gut-derived hormone; the final effect is an accelerated bone loss [4]. Patients affected by secondary hyperparathyroidism should obtain bone benefits from oral supplementation of vitamin D [5,6]. In fact, the European Association for Endoscopic Surgery (EAES) Clinical practice guidelines on bariatric surgery strongly recommend vitamin D supplement post-surgery because the anticipated benefits outweigh the potential risks of vitamin therapy [7].

Poor protein digestion and absorption, secondary to altered biliary and pancreatic function, is involve in protein malnutrition and can be observed after bariatric surgery [4]; albumin levels can be considered as marker of protein deficiency [6].

Low serum levels of fat-soluble vitamins (vitamin A, K and E) usually occur after bariatric procedure [4,5].

As a “treatment gap” option, a novel minimally invasive procedure, i.e., endovascular embolization of the gastric fundus arterial supply (endovascular bariatric surgery -EBS), has been proposed as a supplementary technique with successful results in animal models and a few recently published small clinical trials [8,9,10,11,12,13]. 

We report herein an updated review on the rational, pathophysiology, experimental, and clinical outcomes with this procedure in obese patients. 

## 2. Pathophysiological Basis for EBS

Gastric fundus is mainly supplied by the left gastric artery (LGA) and sometimes by the gastroepiploic artery. The stomach has a neurohumoral role on hunger regulation through ghrelin: this is the rationale for gastric fundus embolization. Ghrelin is a ligand of the growth hormone secretagogue receptor (GHS-R) in neuropeptide Y (NPY) and agouti-related peptide (AgRP) in the arcuate nucleus of the hypothalamus with a downstream effect to inhibit the release of the **α**-melanocyte-stimulating hormone. Therefore, ghrelin acts to increase appetite and food intake, increasing weight gain (Figure 2). Practically, ghrelin plasma level rises sharply shortly before meals, which correlates with hunger sensation that occurs before consuming food. Conversely, ghrelin falls immediately after eating, which correlates with the sense of satiation after eating [14,15]. In addition, ghrelin downregulates anorexigenic hormone receptors for PYY, GLP-1, and cholecystokinin and reduces the sensitivity of gastric distension by selectively inhibition of gastric subpopulation of mechanically sensitive vagal afferent nerves [16,17]. Nearly 90% of body ghrelin is produced in the ghrelin-secreting cells of the stomach, mainly in the gastric fundus with fewer amounts produced in the small intestine, brain, and pancreas. Therefore, the main goal of the EBS procedure is to reduce ghrelin production by the stomach.

It has been previously reported in small retrospective studies [18,19] that patients with gastrointestinal bleeding treated with LGA embolization had significant weight loss after the procedure. Although these studies are difficult to interpret due to the small sample size, associated comorbidities, high-risk profile of the population included, and variable follow-up, they proved the concept and set the basis for gastric fundus embolization as a possible treatment for obesity.

## 3. EBS Procedure

The celiac trunk branches from the aorta at the level of the twelfth thoracic vertebra (T12). The LGA is the first and smaller branch of the celiac trunk, even if there are less common possibilities of independent origins from the aorta, splenic artery, common hepatic artery, gastroduodenal artery and superior mesenteric artery. It runs along the superior portion of the lesser gastric curvature and anastomoses with the right gastric artery that arises from the common hepatic artery. The left gastroepiploic artery (GEA) is the largest branch of the splenic artery and gives gastric branches to both surfaces of the stomach. It anastomoses with the right GEA that arises from the gastroduodenal artery (Figure 3). Normal anatomic variants are frequent and can be present in up to 30% of patients. 

Although there is no standard bariatric embolization procedure, similar approaches have been used in the majority of studies done so far. Through the femoral or radial artery, a selective digital subtraction angiography (usually in AP, LAO 60°, or LAO 90° projections) of the celiac trunk is performed to identify the LGA and other potentials embolization targets (all arteries supplying the gastric fundus and potential accessory gastric arteries); the decision to embolize other vessels, especially the gastroepiploic artery, is based on their contribution to fundal blood supply (assessed angiographically). Different types of diagnostic catheters can be used for this purpose (JR, Simons, Cobra, SOS, among others) depending on the operator’s preference and experience. Cone beam CT has also been used by some authors for best determining fundal perfusion and the eventual need to embolize other arterial territories. After targets have been identified, selective cannulation and angiography using the same catheter or a microcatheter are performed. Embolic material choice has been variable throughout trials with 300–500 μm and 500–700 μm microspheres and 300–500 μm or 500–700 μm polyvinyl alcohol (PVA) particles used. LGA embolization can also be performed with an occlusion balloon microcatheter (OBC) advanced into the target artery over a standard guidewire: a subsequent balloon inflation at the OBC tip can be used to prevent retrograde reflux, with tip pressure/resistance monitored to prevent overembolization and antegrade reflux. Embolization is taken to stasis, which was defined as the visual absence of the flow of contrast after five heartbeats; postembolization DSA is usually acquired to confirm the success of embolization (Figure 4). Unfortunately there are no consensus statements to standardize this procedure. 

## 4. EBS Preclinical Evidence

In 2007, Arepally and coauthors published the first experience of catheter embolization of the LGA to reduce systemic plasma levels of ghrelin in a swine model [20]. The study showed a reduction in ghrelin but no significant weight change. Micro-ulcers at the gastroesophageal junction were also observed in the euthanized animals. A follow-up study with a sham procedure in the control group evaluated ghrelin levels and natural weight gain in 4 weeks follow-up. A significant decrease in weight (7.8%) compared to the control group (15.1%) was shown [21]. 

In a canine model, Bawudun et al. [22] showed a significant decrease in plasma levels of ghrelin, abdominal subcutaneous fat, and body weight in Lipiodol-embolized (or a combination of Lipiodol and polyvinyl-alcohol) animals compared to control group. Peak effect was obtained between weeks three and four and compensatory rise in plasma levels of ghrelin was observed at seven weeks after embolization. According to the author, this result may suggest the compensatory production of ghrelin that occurred in the gastric fundus after the embolization. 

Paxton and colleagues obtained 55% of ghrelin reduction associated with significant weight loss at eight weeks in a swine model using 40 μm microsphere as embolizing agent [23]. The same author evaluated by histology the gastric mucosa of swine after the procedure and demonstrated healed or healing ulcers in 50% of the gastric body of treated animals [24].

In 2016, to assess whether the number of fundal arteries embolized could impact the ghrelin reduction and gastric ulceration rate, the same authors utilized embolic microspheres into four arteries supplying the gastric fundus, vs. two arteries vs. one artery. Only the group of swine undergoing complete embolization demonstrated significant ghrelin reduction compared to sham control animals. Gastric ulcers were present in 50% of animals that embolized four arteries compared to 40% in the other groups [25]. 

In 2017, Kim et al. [26] performed EBS in five swine by selectively infusing 50–150 or 150–250 μm PVA particles into the gastric arteries while five animals were treated with saline infusion as a sham procedure. Endoscopy was performed three weeks after EBS to see whether any gastric ulcer occurred. Celiac trunk angiography was performed eight weeks after EBS [27]. No statistically significant differences in ghrelin levels were observed after eight weeks despite a reduction compared to the control group. In the embolized group, ulcerations were identified in three animals (60%). Re-canalization of the embolized arteries was identified on follow-up angiography in three animals (60%), respectively, suggesting that EBS procedure with PVA particles can transiently suppress ghrelin levels in embolized animals. However, ulcerations of gastric fundus and recanalization of the embolized arteries are present in the long-term follow-up. 

A comparison of the studies involving animal models is reported in Table 1. 

## 5. EBS Human Clinical Evidence

The available clinical evidence on bariatric embolization has been thoroughly described in previous reports [10,27,28]. The first human experience was published in 2013 by Kipshidze and colleagues [11] with its two years follow-up published in 2015. In this five-patient series (mean age 44.7 ± 7.4), the average weight decreased from 128.1 ± 24.4 to 106 ± 21 at two years. All procedures were performed through femoral access using 6F JR 4 catheters for angiography and Excelsior 1018 Microcatheter (Boston Scientific Corp., Cork, Ireland) for selective cannulation. Embolization was done with 300 to 500 μm embospheres (Biocompatibles UK Limited, Surrey, United Kingdom). There were no major procedural complications. Three of the five patients complained of abdominal discomfort after the procedure, all with unremarkable follow-up gastroscopies.

The GET LEAN trial [13] (Gastric Artery Embolization Trial for the Lessening of Appetite Nonsurgically) published in 2016 analyzed the safety and efficacy of LGA embolization at six months. With four patients included (mean age 41 years, range, 30–54), average weight loss was 9.2 kg (range 2.7–17.2) at six months follow-up. Ultrasound (US) guided right femoral or left radial access and 4–5 F Simmons 1 catheter for angiography were use and embolization was performed with 300 to 500 μm microspheres through a microcatheter. No major complications were reported. Three of the five patients complained of mild nausea, occasional vomiting, and mild epigastric discomfort immediately following the procedure that were resolved within 24 h for two patients and within 3–4 days for the other.

In another series published by Bai [8] in 2018 that included five patients with a mean age of 42.8 ± 13.9, weight loss was on average 12.9 ± 14.66 kg at nine months follow-up. Operators used US guided femoral access in all patients, 5F standard catheter for angiography, Progreat 2, 7 F microcatheter (Terumo) for selective cannulation, and embolization was done with 500–710 μm PVA particles (COOK Incorporated, Bloomington). No major complications were reported: four patients experienced slight epigastric discomfort in the first hours after the procedure, which resolved within 48 h. One patient developed a small ulceration (3 mm in length) below the cardia (grade II according to CTCAE v4.0), which endoscopically healed 30 days later after treatment with omeprazole (20 mg).

Pirlet in 2019 [12] studied seven morbid-obese patients (mean age 48 ± 7 years, mean BMI of 52 ± 8 kg/m^2^) who were referred for coronary angiography. Weight loss was on average was 13  ±  17 kg (median loss: −11 kg [0, −25]) at up to 12 months after the procedure. The procedures were done through 5–6F right radial access, and selective cannulation of the LGA was done by 5F JR catheter in all but one patient where a 3F Renegade Microcatheter (Boston Scientific Corp., Cork, Ireland) was used. Embolization with 300–500 μm PV particles (Cook Medical, Bloomington, Indiana, USA) was performed. No major complications were observed in this series. Six patients had transient epigastric discomfort resolved with pro-ton pump inhibitors (PPIs).

In a retrospective series published by Elens and colleagues [9] in 2019 that included 16 patients who underwent embolization of the LGA the mean weight loss was 8 ± 5.12 kg. The procedure was successful in all but one patient. Four patients (25%) were lost in follow-up. Femoral access was used in all patients; celiac trunk angiogram was done with a 5F cobra catheter (Cook Medical, Bloomington, Indiana, USA), LGA angiography with 5F MP catheter, selective cannulation with Progreat 2.7F (Terumo, Tokyo, Japan) microcatheter and embolization was performed with 300–500 μm embospheres (Merit Medical) in the first patient and 500–700 μm embospheres in the remaining patients. The patient treated with 300–500 μm embosphere had gastric ulceration on control endoscopy that resolved at three months. One major complication was reported in this series of a patient that ended in the intensive care unit for pancreatitis, splenic infarct and late gastric perforation.

The BEAT trial [29] (Bariatric Embolization of Arteries for the Treatment of Obesity) 1-year results included 20 patients (mean age of 44 years 6 +/− 11 years) and the procedure was successful in all of them. The mean excess weight loss at one year was 11.5% (95% CI: 6.8%–16%; *p* < 0.001). In this series, patients underwent two-weeks pre-treatment with omeprazole and sucralfate that continued up to 6 weeks post-procedure. Operators used femoral access, celiac trunk angiography plus cone-beam CT to evaluate the gastric arterial distribution. Catheterization was performed with a 5-F SOS guiding catheter (Angiodynamics, Latham, NY), and a 2.9-F high-flow microcatheter (Maestro; Merit Medical) was utilized for selective cannulation. After administration of 200 mg of nitroglycerin and 2.5 mg of verapamil (only into the LGA), embolization was performed with 300–500 μm embospheres (Merit Medical). No major complications were reported on this trial. Eleven minor adverse events occurred in eight participants. One participant had subclinical pancreatitis, evident by transient elevation of lipase levels during the hospital stay that was treated with supportive care and discharged within 48 h, with no further clinical sequels.

Zaitoun and collaborators recently published a pilot study of EBS in 10 patients with obesity (BMI > 30 kg/m^2^) and prediabetes (hemoglobin A1c (HbA1c) level 5.7%–6.4%) [30]. A statistically significant reductions in HbA1c (from 6.1% ± 0.2 to 4.7% ± 0.6; *p* < 0.0001), mean body weight (from 107.4 kg ± 12.8 to 98 kg ± 11.6; *p* < 0.0001), and mean BMI (from 37.4 kg/m^2^ ± 3.3 to 34.1 kg/m^2^ ± 3; *p* < 0.0001) was observed at 6 months follow-up. Embolization was performed with 300–500 μm microspheres. The authors also reported a significant positive correlation between BMI and HbA1c levels (r = 0.91; CI, 0.66–0.98; *p* = 0.0002).

In 2020 the LOSEIT group, led by Vivek Reddy of the Icahn School of Medicine at Mount Sinai in New York City, published the first in-human, sham-controlled, randomized clinical trial about EBS [31]. In this trial 40 patients (BMI of 35.0 to 55.0 kg/m^2^ and age 21 to 60 years) underwent randomization in 1:1 to either sham or transcatheter bariatric embolotherapy (TBE) targeting the left gastric artery; patients randomized to sham were unblinded at 6 months and crossover to TBE was allowed. Patients of both groups received a lifestyle support (diet and behavioral education for weight loss). TBE has been performed via femoral arterial access using standard 6-F guiding catheters for celiac artery angiography. In a case the gastroduodenal artery was the origin of the left gastric artery which supplied the gastric fundus and in one patient the embolization target vessel was the left hepatic artery. An occlusion balloon microcatheter (Endo-bar Solutions LLC, Orangeburg, New York) was advanced into the target artery and a balloon at the tip was inflated to prevent retrograde reflux. The embolization was achieved by using 300- to 500-mm microspheres (BeadBlock, Biocompatibles Ltd., Farnham, United Kingdom) into the LGA. The procedure was repeated until adequate angiographic stasis was achieved with the balloon deflated over 5 cardiac cycles. Patients randomized to sham received propofol only, without arterial access. At 6 months, in both the intention-to-treat and per-protocol populations, the total body weight loss was greater with TBE (7.4 kg/6.4% and 9.4 kg/8.3% loss, respectively) than sham (3.0 kg/2.8% and 1.9 kg/1.8%, respectively; p 1⁄4 0.034/0.052 and p 1⁄4 0.0002/0.0011, respectively); the total body weight loss was maintained with TBE at 12 months (intention-to-treat 7.8 kg/6.5% loss, per-protocol 9.3 kg/9.3% loss; p 1⁄4 0.0011/0.0008, p 1⁄4 0.0005/0.0005, respectively). After 1 week, all the patients underwent endoscopic examination: 5 cases of asymptomatic ulceration in the treatment group occurred (4 small superficial ulcers in the sub-cardiac region of the stomach and 1 superficial ulcer in the greater curvature).

Bariatric Embolization of Arteries With Imaging Visible Embolics (BEATLES) (BAE2) is an on-going, randomized, sham-controlled study sponsored by Johns Hopkins University [32]. The aim of the trial, with an estimated enrollment of 59 participants, is to evaluate the change in body weight 12 months after randomization in the bariatric embolization procedure arm versus the control (sham) arm; the estimated study completion date is December 2023.

A comparison of the studies involving humans is reported in Table 2.

## 6. Current and Future Perspectives

Considering the huge impact of obesity in public health worldwide, novel treatments are of utmost importance to reduce associated morbidity and mortality related to this condition and its complications. Early experimental and initial clinical data have suggested that BES may be a promising technique to reduce weight and hormonal imbalance in the obese population. Compared to bariatric surgery, BSE seems to be less invasive, well-tolerated, with shorter recovery time, and is probably cost-saving; further randomized clinical trials are needed to confirm this hypothesis and to evaluate eventual consequences regarding metabolic and nutritional status of patients underwent EBS. The duration of BES procedure varies from 80 to 100 min, with a fluoroscopy time of 32 ± 14 min. Current human clinical evidence shows that bariatric LGA embolization is an effective treatment that is associated with statistically significant weight loss during follow-up. In addition to weight change, some clinical series have also demonstrated a reduction in serum level of ghrelin and/or leptin [8,11,29,31] and in the level of hemoglobin A1c [29,30] and mean total cholesterol [29]; a statistically significant mean decrease in diastolic blood pressure in the embolization arm was observed in one trial [31]. Furthermore, the procedure is associated with improvements in quality-of-life after embolization, especially regarding the domains of physical function, self-esteem, sexual life, and public distress. The most frequent side effects after the procedure are nausea, vomiting and epigastric pain, and minor complications such as transient superficial mucosal ulcers are common after LGA embolization (usually treated with proton pump inhibitors and spontaneous healing within 4 weeks to 3 months without the needed of further hospitalization); only one case of major complications after the procedure (pancreatitis, splenic infarct and late gastric perforation) was reported by Elens et al. [9].

Distal embolizing agents (mostly embospheres and PVA particles) have been the main embolic material used in recent trials. Although bariatric embolization seems to be safe in this initial phase and results in terms of weight loss are promising, we need further data on patients subsequently treated with conventional bariatric surgery (given that the gastroesophageal junction is de-vascularized) to establish if BES can be utilized as bridge-therapy to surgery in this population. The ideal embolic material remains to be elucidated to achieve what seems to be a perfect safety-efficacy balance mostly on the long term. A larger well designed RCT ideally with a sham-controlled group is eagerly expected to establish the future role of BES in favoring weight loss and hormone homeostasis and for prevention of obese-related morbidity and overall mortality. There is also the needed of a trial of comparison between BES and optimal medical therapy, including new drugs like Semaglutide [33]. At last there are not data of comparison between BES and surgical techniques to evaluate if BES should be considered as an alternative in those patients who are not passible of surgery (multimorbid patients).

## Figures and Tables

**Figure 1 nutrients-13-02541-f001:**
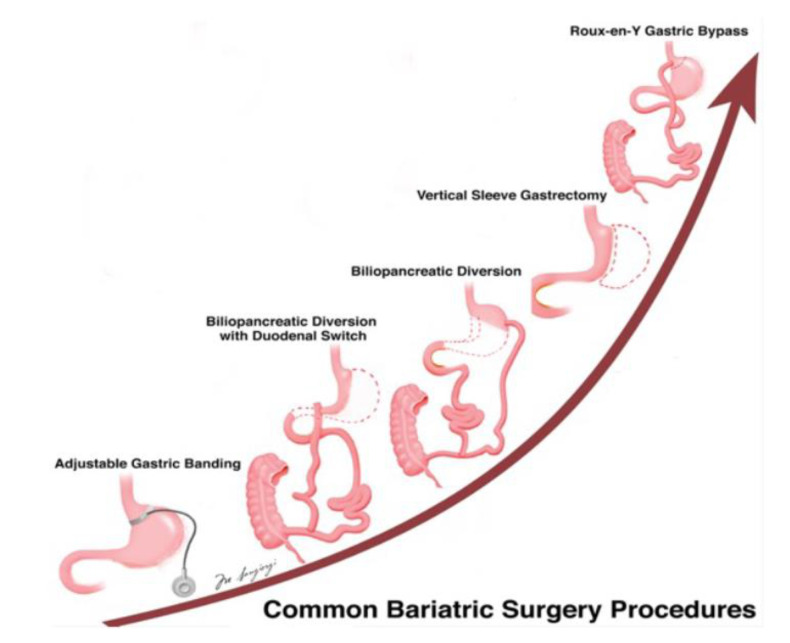
Overview of different bariatric surgery options. The most effective options in weight reduction are the Roux-en-Y gastric bypass and sleeve gastrectomy. Arrow indicates an increase in efficacy.

**Figure 2 nutrients-13-02541-f002:**
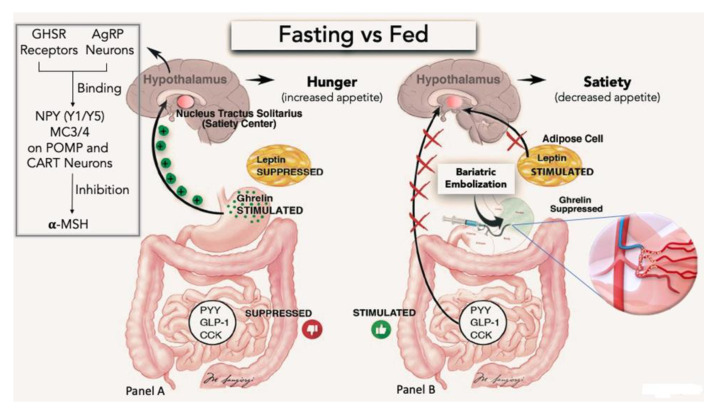
Hormonal changes and diagram of the ghrelin signal pathway. (**A**) The “hunger hormone” ghrelin is secreted by the gastric fundus, whereas peptide YY (PPY), cholecystokinin (CCK), and glucagon-like peptide (GLP-1) are secreted in the gut from L cells. Adipocytes produce leptin (LPT). In the fasting state, decreased food intake suppresses the release of PPY, GLP-1, CCK, and LPT, while stimulates ghrelin production from the stomach. Ghrelin binds in the hypothalamic arcuate nucleus to growth hormone secretagogue receptor (GHSR) in neuropeptide Y (NPY) and agouti-related peptide (AgRP) neu-rons. NPY and AgRP bind subsequently to NPY subtype 1 and 5 (NPH Y1/Y5) and melacortin-3 and-4 (MC3/4) receptors on proopiomelanocortin (POMP) and cocaine-amphetamine-regulated transcript neurons (CART), inhibiting the release of **α**- melanocyte-stimulating hormone (**α**-MSH). By inhibiting **α**-MSH, ghrelin acts to increase hunger and food intake. (**B**) BES procedure reducing ghrelin production in the stomach fundus area, mimics a fed state characterized by PYY, GLP-1, CCK, and LPT hormone increases. As a result, appetite decreases and an increase in the feeling of satiety occurs.

**Figure 3 nutrients-13-02541-f003:**
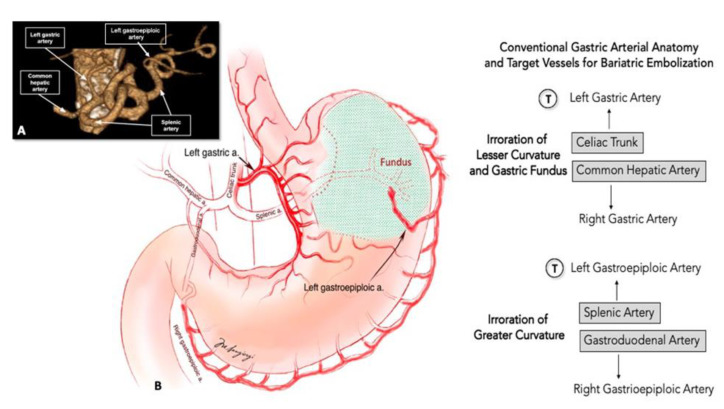
Voxel Gradient Angio-CT 3D reconstruction (**A**) and schematic drawn (**B**) of left gastric artery and left gastroepiploic artery normal anatomy. Most commonly the left gastric artery originates from the celiac trunk. Less frequently, the artery may arise directly from the aorta, splenic artery, common hepatic artery, and superior mesenteric artery. The superior part of the greater curvature of the stomach is supplied by the left gastroepiploic artery while the inferior part of the greater curvature by the right gastroepiploic artery. T indicates target for BES procedure.

**Figure 4 nutrients-13-02541-f004:**
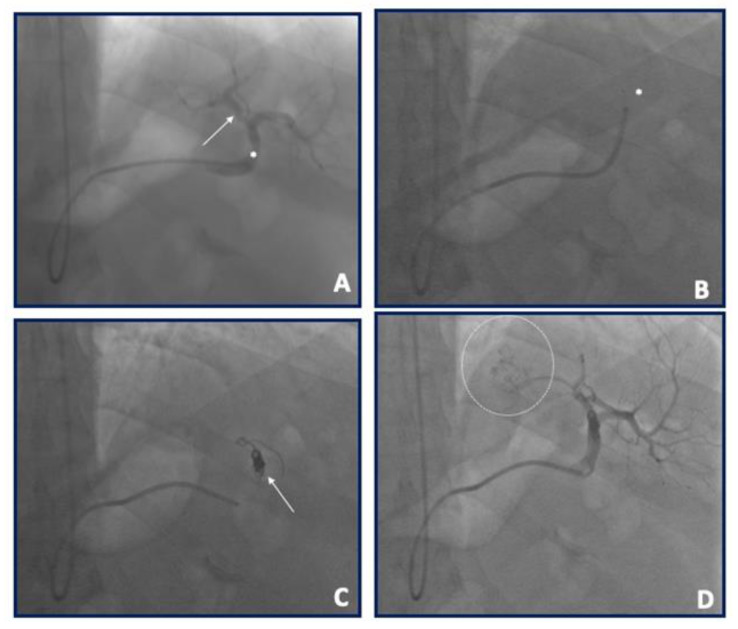
Case example of a left gastroepiploic artery (arrow) embolization in a 55 years old male with a BMI of 43.2. (**A**) Selective angiography by multipurpose 6F 125 cm into the splenic artery (asterisk). (**B**): A Rebar 0.27′ microcatheter (Medtronic, Santa Rosa, CA, USA—asterisk) was advanced into the multipurpose catheter selectively engaging the gastroepiploic artery. (**C**) Three contour spirals (Medtronic, Santa Rosa, CA, USA) of different size and length (4 × 40 mm; 4 × 40 mm; 5 × 30 mm) were subsequently released (arrow). (**D**) Final angiography with target fundus zone indicated by dashed circle.

**Table 1 nutrients-13-02541-t001:** Experimental studies evaluating BES procedure for weight reduction.

Author/Year/REF.	Animal Model	N of Animals	Follow-Up	Absolute Weight Change (%)	Absolute Ghrelin Change (%)
Arepally 2007 [20]	Swine	5 2 CTRL 4 LDSM 4 HDSM	4 weeks	8.6 increase 1.4 increase 1.4 increase	15% increase 130% increase 60% decrease
Arepally 2008 [21]	Swine	10 5 CTRL 5 SM	4 weeks	15.8 increase 7.8 increase	10.1 increase 12.9 decrease
Bawudun 2012 [22]	Dog	15 5 CTRL 5 EO 5 PVA	8 weeks	5 increase 5 decrease 10 decrease	13.6 increase 15.8 decrease 30.1 decrease
Paxton 2013 [23]	Swine	12 6 CTRL 6 MS	8 weeks	28 increase 10 increase	30 increase 30 decrease
Paxton 2014 [24]	Swine	12 6 CTRL 6 EO	8 weeks	Not Reported Not Reported	Not Reported Not Reported
Paxton 2016 [25]	Swine	22 6 CTRL 6 EA [8]5 EA [2]5 EA [1]	4–8 weeks	26.5 increase 8.7 increase 19.7 increase 23.0 increase	19.9 increase 29.9 decrease 18.2 increase 234 increase
Kim 2017 [26]	Swine	10 5 CTRL 5 PVA	2 mos	30% increase 27% increase	No change 40% decrease

CTRL = control; LDSM = low dose sodium morrhuate; HDSM = high dose sodium morrhuate; EO = ethiodized oil (Lipiodol); PVA= polyvinyl-alcohol; MS = microspheres; EA (*n*) = number of embolized arteries.

**Table 2 nutrients-13-02541-t002:** Clinical Studies Evaluating BES procedure for Weight Reduction.

Author/Year/REF	Study Design	N of pts	Follow-UpLength	Total Weight Loss (%)	Embolic Material
Kipshidze 2015 [11]	Prospective	5	2 years	22 kg (17.1%)	300–500 μm BeadBlock embospheres
GET-LEAN 2016 [13]	Prospective	4	6 months	9.2 kg (8.5%)	300–500 μm BeadBlock embospheres
Bai 2018 [8]	Prospective	5	9 months	12.9 kg (12.6%)	500–710 μm PVA particles
Pirlet 2019 [12]	Prospective	7	12 months	13 kg (4.7%)	300–500 μm PVA particles
Elens 2019 [9]	Prospective	16	12 months	8 kg (10%)	500–700 μm Merit Medical embospheres
BEAT 2019 [29]	Prospective	20	12 months	7.6 kg (11.5%)	300–500 μm Merit Medical embospheres
Zaitun 2019 [30]	Prospective (pre-diabetic)	10	6 months	9 kg (8.9%)	300–500 μm Merit Medical embospheres
LOSEIT 2020 [31]	Randomized, sham-controlled	40	12 months	9.3 Kg (9.3%)	300–500 μm BeadBlock microspheres
BEATLES 2023 [32]On-going Trial	Randomized, sham-controlled	56	12 months	On-going trial	100–200 µm radiopaque microspheres

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
