# Peer review of "Endovascular Bariatric Surgery as Novel Minimally Invasive Technique for Weight Management in the Morbidly Obese: Review of the Literature"

_nutrients, 2021, doi:10.3390/nu13082541_

Round 1

Reviewer 1 Report

Abstract: Overall this section reads well with only some small grammar issues I have listed below. However, what I would like to see changed is the lack of mention made to MIS methods for bariatric surgery. Open surgery is by far not the norm in Bariatric surgery and as a result is probably not the most apt comparison when bringing up endovascular bariatric surgery. MIS needs to be included in this opening section.

P1L14 are of “the” utmost importance.

~

Introduction: I think this section is well written and to the point. I like the summary they gave of the proposed mechanism for endovascular bariatric surgery with gastric fundus embolization. Below are some word choices and clarifications that need to be made. However, overall this section is in good shape.

P2L44/45 consents should probably say “results in” or something similar. P2L49 says has “limited adherence” which I think the authors mean that patients are not consistent in taking the medication, this phrasing needs to be clarified. P2L58 I think division here should say diversion?

~

Figure 1: This figure is easy to understand but I would specify in the description what ‘most effective’ means. Is this related to weight loss, resolution of co-morbidities? I’m surprised to see Duodenal switch being so low when you talk about effective. I don’t think the figure needs to be revised but the description lacks clarity.

~

Figure 2: Looks good, no changes here.

~

EBS Procedure: This section does a good job of quickly summarizing the anatomy pertinent to this procedure. It also does a good job of presenting the basic tenants of the endovascular bariatric procedure. I would like to see some mention of any consensus statements or projects to standardize this procedure if they are available. If not a statement should be made that there is no standardization at this time.

~

Figure 3: A few words in this section have hyphens that don’t need them. The content of the figure appears to be accurate and well presented. However, this may be my own ignorance, but what is “irroration” I don’t see an explanation of what this means and couldn’t find it with a quick Google search.

~

Figure 4: Well presented figure, no changes

~

EBS Preclinical Evidence: On P6L168-169 the authors state in passing that there was a compensatory rise in plasma levels of ghrelin at 7 weeks post embolization. Can you expand on that more? Did the authors of that study have a hypothesis on why this happened and if it’s important? I’d like to see a bit more about this as it seems like an important fact to better discuss. P6L178-180 the authors commend on gastric ulcers present after embolization. I’d like to see a little discussion on why this is important again as it would seem like a problem if these patients all had bleeding gastric ulcers after the procedure. In summary this section does a good job of reviewing the evidence surrounding EBS but I would like to see more expansion on the ideas listed previously to really bulk up the section. At the moment some of it reads like a list of data without much context.

~

Table 1: Good summary, no changes

~

EBS Human Clinical Evidence: This section is well written and easily summarizes the available human clinical evidence related to EBS. I would make very minimal changes to grammar listed below.

P7L199 says “in man experience”, would sound better as “human experience”. P8L227 late should be “later”

~

Table 2: Well presented, no obvious issues

~

Current and future perspectives: The authors do a good job of succinctly summarizing the desirable qualities of EBS but I think more can be done to talk about who this procedure would be used for. In the author’s estimation do they see this as a bridge to surgery? Or do they see it as a primary procedure? It’s not clear at this time who this would be useful for as there are very real risks to the GE jxn and gastric pouch if the patient were to undergo a eventual bariatric surgery. This should not be glossed over and should be expanded in this section.

P10L329 “Despite” doesn’t make sense to start this sentence. The authors need to clarify what they are saying in this sentence to make it more clear.

~

The following words do not need hyphens: P1L3 technique, P1L14 increases, P1L18 surgery, P1L21 bariatric, P1L22 follow-up, P1L36 occur, P1L40 importance, P2L53 convention, P2L58 banding, P2L59 individuals, P2L60 uncontrolled, P2L61 modifications, P2L71 clinical, P3L77 gastroepiploic, P2L91 stomach, P3L94 although, P4L113 include, P4L114 mesenteric, P4L123 performed, P4L125 artery, P4L132 angiography, P4L140 postembolization, P6L185 significant, P6L191 arteries, P7L200 colleagues, P7L224 patient, P8L225 accodring, P8L226 endoscopically, P8L228 published, P8L233 embolization, P8L234 complications, P8L236 proton, P8L238 patients, P8L247 intensive, P8L249 results, P8L252 continued, P8L261 discharged, P8L265 significant, P8L274 underwent, P8L275 targeting, P9L278 standard, P9L291 respectively, P10L311 promising, P10L312 bariatric, P10L314 treatment, P10L318 statistically, P10L320 embolization, P10L321 distress, P10L323 embolization, P10L333 material, P10L337 homeostasis.

Reviewer 2 Report

The manuscript describes an interesting new technique, even if it is perhaps not quite ready for everyday clinical use. The review is technically well done. 

Author Response

Thank you.

Reviewer 3 Report

Dear colleagues, 

Thank you for giving me the opportunity to review this very interesting and relevant paper!

Introduction part is very long.  The meaning of obesity in general is absolutely clear and can be mentioned in just few sentences. Please keep the introduction substantially shorter.

Pointing out the drugs, I am thinking that the brand-new Semaglutid is to be mentioned as well. This drug promises up to 14% EWL, published this year in British Journal of medicine.  This paper can be also mentioned in the discussion part.

Line 53- 55: The complications after endoscopic approaches are very rare, so I do not think it is fair to to display it like this.

Line 63- 64: The described complications of bariatric surgical procedures are very rare in Center- hospitals and they are investigated by very large numbers of patients.  The safety of bariatric surgery is already proven and every new intervention needs to show similar results in comparable patient numbers.  Please pay attention to this, throughout your whole presentation (also discussion).

Figure 1 is absolutely unnecessary. Please remove it. 

Inside the animal- trials, gastric ulcera seem to be a relevant problem. Why is that? Please try to give some explanation. 

The presentation of the human- Study data has to be made with uniform Weight parameters like EWL% or TWL%. Table 2: average weight of the group and EWL/ TWL. This would give much better overview and understanding of the data. 

Line 312- 313: The conclusion EBS is less invasive and patients have shorter recovery can not be made like this, before larger trials about it have occur. Please change this. Display the average time of the EBS procedure. RYGB in centers needs about 60- 90 Min, SG less than 1 hour. 

"Current and future perspectives" needs to bring more in comparison of EBS and the other techniques. 

Drugs like Semaglutide, gastric ballon, standards like RYGB & SG. 

Please show some possible disadvantages of EBS: Gastroparesis after intervention (Nausea, Vomiting etc.). Ulcera?

Describe wich clinical field can be the best for EBS- multimorbide patients? low BMI patients? 
